# Introducing CAR-T Therapy in Kazakhstan: Establishing Academic-Scale Lentiviral Vector and CAR-T Cell Production

**DOI:** 10.3390/biom15081166

**Published:** 2025-08-14

**Authors:** Viktoriya Keyer, Aitolkyn Kydyrbayeva, Tolganay Kulatay, Gulzat Zauatbayeva, Dmitrii Bazhenov, Bakytkali Ingirbay, Zhanar Shakhmanova, Maral Zhumabekova, Madina Ospanova, Alexandr V. Shustov

**Affiliations:** 1National Center for Biotechnology, Korgalzhin hwy 13/5, Astana 010000, Kazakhstan; keer@biocenter.kz (V.K.); aitolkyn.yk@gmail.com (A.K.); kulatay@biocenter.kz (T.K.); zauatbaeva@biocenter.kz (G.Z.); bazhenov@biocenter.kz (D.B.); ingirbay@biocenter.kz (B.I.); zhanar.shakhmanova@bk.ru (Z.S.); zhumabekova@biocenter.kz (M.Z.); 2Scientific-Production Center of Transfusiology, Zhanibek Khandar St., 10, Astana 010000, Kazakhstan; sk_ospanova_me@mail.ru

**Keywords:** CAR-T therapy, academic format, point-of-care, CliniMACS Prodigy, hospital exemption, developing economy

## Abstract

CAR-T cell therapy represents a breakthrough in cancer treatment, yet its implementation in developing countries remains challenging due to technical and infrastructural barriers. This study aimed to establish clinical-scale CAR-T production in Kazakhstan, a country with no prior experience in advanced cell and gene therapies. We implemented a complete CAR-T manufacturing pipeline, including in-house lentiviral vector (LV) production and automated CAR-T cell processing using the CliniMACS Prodigy system. Two anti-CD19 CAR LVs were used, one modeled after FDA-approved Kymriah (4-1BB costimulation) and another replicating Yescarta (CD28 costimulation). The vector produced locally achieved functional titers of 1.5 × 10^10^ TU/mL after concentration. Twelve clinical-scale CAR-T products were manufactured, exhibiting a memory-skewed T-cell phenotype. Functional assessments revealed that CD28-based CAR-T cells produced significantly higher Th1 cytokines (IFN-γ, TNF-α, IL-2; *p* < 0.05) than 4-1BB-based cells, though both demonstrated comparable cytotoxicity against CD19+ targets. These findings demonstrate the feasibility of establishing CAR-T production in resource-limited settings using a decentralized manufacturing framework. This work provides a scalable model of CAR-T therapy production in developing regions, suitable for clinical implementation using the hospital exemption framework. Critical gaps in access to advanced immunotherapies, including CAR-T, in the Central Eurasia region are addressed.

## 1. Introduction

Chimeric antigen receptor (CAR) T-cell therapy represents a groundbreaking advancement in immunotherapy, wherein autologous T-cells are genetically modified to express synthetic receptors (CARs) that enable specific recognition of tumor-associated antigens. This engineered targeting mechanism promotes potent cytotoxicity and tumor elimination [1,2,3]. CAR-T therapy has gained regulatory approval for treating hematologic malignancies, including B-cell leukemias, lymphomas, and multiple myeloma, showcasing unprecedented clinical efficacy in relapsed or refractory (r/r) cases where conventional therapies fail [4,5].

The first FDA-approved CAR-T therapy, Kymriah (tisagenleucel), was commercialized by Novartis in 2017, followed by others including Yescarta, Tecartus, Breyanzi, Abecma, and Carvykti [6,7]. Despite their transformative clinical outcomes, these therapies are among the most expensive cancer treatments, with prices exceeding $300,000 per infusion. This prohibitive cost severely limits accessibility, particularly in low- and middle-income countries [8,9].

The introduction of CAR-T therapy into clinical practice in developing countries, such as the authors’ country, Kazakhstan, where CAR-T is currently unavailable, is eagerly awaited, given its potential to treat otherwise incurable cancers. However, its implementation faces significant challenges, including technological complexity, high costs, and infrastructural limitations [10,11]. Successful examples of clinical-scale CAR-T production in resource-constrained settings [12,13,14,15] underscore the need to disseminate protocols and collaborative training to empower hospitals and biotech laboratories in the developing world.

Globally, CAR-T therapy is introduced into new markets via two predominant strategies: the industrial and academic models. The industrial approach relies on centralized, large-scale manufacturing by commercial providers (e.g., Novartis and Gilead), delivering standardized products to multiple regions. Conversely, the academic model enables decentralized production; qualified clinical centers manufacture CAR-T cells for on-site patient use, or biotech labs produce CAR-T cells only for their collaborating hospitals, often at reduced costs [16]. This latter approach is particularly relevant for resource-limited settings, where affordability and regional adaptability are critical [17].

To date, all clinically approved CAR-T cell products rely on viral transduction using lentiviral (LV) or gamma-retroviral (gRV) vectors, to deliver the CAR gene into patient-derived T-cells. However, the global shortage of clinical-grade viral vectors remains a critical bottleneck for the expansion CAR-T programs, complicating efforts to expand access to this therapy [18]. Securing reliable vector supply chains is thus essential for sustainable CAR-T implementation [19].

This article presents the authors’ experience with establishing clinical-scale anti-CD19 CAR-T cell production in Kazakhstan, an upper-middle-income country with 20 million people, and a high unmet need in CAR-T. We established local lentiviral vector (LV) production, achieving yields of approximately 10^10^ transduction units (TU) per batch. We then implemented decentralized manufacturing using the CliniMACS Prodigy platform, an advanced cell processing system from Miltenyi Biotec with the capacity to produce clinical-scale CAR-T cell products and the convenience of performing all manufacturing steps in a closed system with minimal open handling [20].

The experience described in this study may assist biotechnology laboratories and interested hospitals in developing countries in accelerating their manufacturing readiness for CAR-T therapy production.

## 2. Materials and Methods

### 2.1. CAR Designs and Lentiviral Vectors

Two chimeric antigen receptor constructs (CARs) used in this study, designated CAR.TM8-BBz and CAR.TM28-28z, belong to the second-generation CARs, which are characterized by the presence of a single intracellular costimulatory signaling domain, along with the CD3ζ activation domain. The two used CARs are anti-CD19 CARs, but they differ in their molecular architecture, particularly in the signal peptide, hinge region, transmembrane domain, costimulatory domain, and interdomain linkers. Figure 1a shows schematic representations of the CAR constructs, while Figure 1b presents the comparison of amino acid sequences.

The gene-encoding CAR.TM8-BBz (CD8 hinge/transmembrane, 4-1BB costimulatory domain, and CD3ζ signaling domain) was assembled by the authors to replicate the published molecular structure of CTL019, which is reportedly the CAR used in the registered therapy Tisagenlecleucel (marketed as Kymriah by Novartis) [2]. The corresponding LV is referred to NCB.LV.CD19-CAR (or V1, for brevity) herein.

The other CAR, termed CAR.TM28-28z, mimics the CAR construct used in Axicabtagene ciloleucel (marketed as Yescarta by Kite Pharma (Los Angeles, CA, USA)) and contains the CD28-derived hinge, transmembrane, and costimulatory domains, along with the CD3ζ signaling domain [21]. The corresponding LV is EFS.LV.CD19-CAR (or V2, for brevity). This vector was kindly provided as a packaged LV preparation by Dr. Emil Bulatov from Kazan Federal University (Kazan, Russian Federation).

Both CAR.TM8-BBz and CAR.TM28-28z receptors target the CD19 antigen, which is specific to B-lineage cells, and contain an antigen recognition domain (ARD) derived from the variable regions of the light (VL) and heavy (VH) chains of the single-chain variable fragment (scFv) FMC63 [22]. However, the VL–VH linkers within the ARD differ between the two constructs (Figure 1b).

The two vectors, NCB.LV.CD19-CAR (V1) and EFS.LV.CD19-CAR (V2), are similar in that both carry the CAR gene controlled via the same EF1α promoter (Figure 1c). These vectors also share identical nucleotide sequences in key regulatory elements, including the primer binding site (PBS), packaging signal (Ψ/Psi), Rev response element (RRE), central polypurine tract/central termination sequence (cPPT/CTS), and 3′ polypurine tract (3′PPT). The 3′ long terminal repeats (LTRs) region is self-inactivating (SIN) with ΔU3 deletion, which eliminates promoter activity of the U3 region in the integrated provirus.

### 2.2. Cell Line for LV Production

HEK293FT cells (ThermoFisher (Waltham, MA, USA), Cat. R70007) were cultured in a complete medium, which was Dulbecco’s Modified Eagle Medium (DMEM) with high glucose (Gibco, Waltham, MA, USA), supplemented with 10% fetal bovine serum (FBS; Gibco, USA), 1% penicillin–streptomycin, 1% non-essential amino acids (NEAAs), and 2 mM of L-glutamine (all from Thermo Scientific, Waltham, MA, USA).

### 2.3. Production of Packaged Vector NCB.LV.CD19-CAR

Figure 2a outlines the LV production workflow. The packaged vector V1 was generated by transfecting HEK293FT cells with the transfer plasmid pNCB.LV.CD19-CAR, alongside helper plasmids: packaging helper psPAX2 (Addgene #12260) and envelope plasmid pMD2.G (Addgene #12259). All plasmids were isolated from transformed DH5α strains via alkaline lysis and purified for transfection through CsCl gradient ultracentrifugation [23]. HEK293FT cells were seeded at density (100,000 cells/cm^2^) in multilayer cell culture flasks (VWR #734-3418). Transfection was performed using calcium phosphate precipitation. One hour prior to transfection, the culture medium was replaced with serum-free medium (complete medium without FBS). Packaging plasmids were combined at equimolar ratios. For each 5-layer flask, 450 µg of plasmid DNA was mixed with 0.25 M CaCl_2_ to a final volume of 7.5 mL. The 2× HeBS buffer (pH 7.12) has the following composition: 274 mM of NaCl, 10 mM of KCl, 15 mM of glucose, 42 mM of HEPES, and 1.4 mM of sodium phosphate. The DNA-CaCl_2_ solution was added dropwise with constant vortexing to an equal volume of 2× HeBS buffer. After 5 min of incubation at RT to allow CaPi formation, the precipitate-containing mixture was evenly distributed in the medium in the culture flask. Cells were maintained in serum-free medium with CaPi for 1 hr before the addition of FBS to a 10% final concentration.

Twenty-four hours later, the medium was replaced with fresh complete medium (100 mL/flask), marking the end of transfection (designated as time zero). At 48 h post-transfection (hpt), the conditioned medium was collected, clarified through low-speed centrifugation (300× *g*, 10 min), and sterile-filtered through 0.45 µM PVDF membranes (Millipore (Billerica, MA, USA), Cat. SLGPR33RS).

During production process optimization, we used the control vector LV/CAR-GFP (Figure 2b–e) in experiments to control transduction efficiency and determine the optimal centrifugation speed for pelleting LV particles. In the control vector, the CAR gene product is fused to green fluorescent protein (GFP), facilitating titer determination. To identify the optimal centrifugation speed, aliquots of the LV/CAR-GFP preparation in thick-walled 1.5 mL tubes (Beckman Coulter (Brea, CA, USA), Cat. 357448) were centrifuged at various speeds (10,000–70,000× *g*) for 2 h at 4 °C (rotor TLA-55, Optima MAX-TL ultracentrifuge, Beckman Coulter).

During batch production runs, LV particles were concentrated via tangential flow filtration (TFF) using a Minimate TFF capsule with a 300 kDa Omega membrane (Pall Life Sciences (New York, NY, USA), Cat. OA300C12) connected to a Tanfil-100 system (Rocker Scientific Co., Taiwan).

Typically, virus-containing medium collected from six multilayer cell culture flasks (~600 mL total) was processed through TFF and concentrated to approximately 30 mL. The concentrate was transferred to 38.5 mL ultracentrifuge tubes (Beckman, Cat. 344058) and further used for LV concentration via ultracentrifugation using an SW28 rotor (Beckman Coulter). The NCB.LV.CD19-CAR vector was concentrated via ultracentrifugation under optimized conditions (20,000× *g*, 2 h). LV particles were resuspended in RPMI 1640 medium (VWR (Radnor, PA, USA), Cat. 392-0429) supplemented with 10% human serum albumin (HSA, Biopharma Plasma LLC, Ukraine) and 0.0001% Pluronic F68 (Sigma (Kawasaki, Japan) Cat. P5556) and stored at −80 °C.

### 2.4. Lentiviral Vector Functional Titer Determination

The physical titer of LV particles was determined by quantifying the p24 capsid protein using the Lenti-X GoStix Plus kit (Takara (Osaka, Japan), Cat. #631280). Samples exceeding the assay’s linear range (>3000 GoStix Values, GV) were diluted at 1:100 or 1:1000. The reported GV values are adjusted by multiplying by the dilution factor.

To determine functional titers, HEK293FT cells were transduced with LV preparations and GFP-expressing (for the control vector) or CAR-expressing cells were counted using flow cytometry. HEK293FT cells were seeded at 4 × 10^5^ cells/well in 6-well plates. Dilutions of LV were prepared (first dilution 1/20 and then with step 1/10) in complete medium with an added 8 μg/mL polybrene (Sigma H9268). The existing medium was aspirated and replaced with 2 mL portions of diluted LV preparations. After 24 h incubation at 37 °C, viral inocula were replaced with fresh complete medium. Cells were harvested 48 hr post-transduction using trypsin/EDTA, washed with PBS, and resuspended in 100 μL of MACSQuant Running Buffer (RB, Miltenyi Biotec Cat. 130-092-747). For CAR detection, cells were stained with 2 μL of biotinylated CD19 CAR Detection Reagent (Miltenyi Biotec 130-129-550) per 1 × 10^6^ cells. The cells were incubated with the CAR-detection reagent for 10 min at room temperature (RT), followed by the addition of 1 mL of RB and pelleting cells (300× *g*, 5 min). After that, 2 μL of anti-biotin-PE antibody (Miltenyi Biotec 130-113-291) was added, followed by incubation (10 min, RT, in the dark) and one more wash in 1 mL RB, as described. The resulting cell pellet was resuspended in 1 mL of RB and used for flow cytometry. Transduction efficiency was quantified via PE positivity using a MACSQuant 10 flow cytometer (Miltenyi Biotec, Bergisch Gladbach, Germany), and the total number of cells was determined using a cell counter. Transducing units (TUs) per milliliter were calculated using the following formula:TU/mL = (% of PE-positive cells/100) × cell concentration (cells/mL) × dilution factor (D).

### 2.5. Source of Starting Material for CAR-T Cell Production

CAR-T cells were manufactured from frozen apheresis products obtained from the Scientific and Production Center of Transfusiology (SPCT, Astana, Kazakhstan). All donors provided written, informed consent prior to admittance to the apheresis procedure. These apheresis procedures were performed for medical indications unrelated to this study. Leukocyte-enriched fractions were collected in SPCT using the Spectra Optia System (Terumo BCT) and subsequently cryopreserved.

Frozen leukapheresis products that remained unclaimed and unused for clinical purposes were transferred to the National Center for Biotechnology (NCB, Astana, Kazakhstan) for use in this study. To protect donor confidentiality, all samples were anonymized, and re-identification is not possible without access to SPCT’s internal database. The study protocol was reviewed and approved by the Institutional Ethics Committee of NCB. A total of twelve apheresis products were utilized. Demographic data and the diagnoses of patients who underwent leukapheresis are listed in Appendix A.

### 2.6. CAR-T Cell Manufacturing

The manufacturing process is schematically depicted in Figure 3.

The total duration of the process was 12 days. On Day 1, a cryopreserved leukapheresis product was thawed and immediately diluted in pre-warmed TexMACS Medium (Miltenyi Biotec (Bergisch Gladbach, Germany), Cat. 130-097-196), supplemented with 1.6 mM of MgCl_2_ and 50 U/mL of nuclease (Benzonase, Merck (Rahway, NJ, USA), Cat. 70664), without cytokine supplementation. After a 15 min incubation at 37 °C, cryoprotectants, such as DMSO, were removed via centrifugation at 300× *g* for 5 min. The resulting cell pellet was resuspended in TexMACS Medium. At this point, aliquots were taken for ancillary procedures, including immunophenotyping and B-cell isolation. The remaining cells were diluted in warmed TexMACS Medium supplemented with 3% human serum albumin (HSA; registered medicinal product RK-LS-5#004526, Biopharma Plasma LLC, Kyiv, Ukraine), 1000–1500 U/mL of recombinant human IL-7 (Miltenyi Biotec, Cat. 170-076-111), and 300–450 U/mL of recombinant human IL-15 (Miltenyi Biotec, Cat. 170-076-186). The cell suspension was transferred to a tissue culture roller bottle for suspension cultures (Biofil, Cat. TCB001005/01.TC.011.0046) and placed in a CO_2_ incubator (model WCI-1200; Wiggens China Co., Ltd., Beijing, China) equipped with a roller installation. The roller apparatus was set to the minimum rotation speed (0.08 rpm), and cells were maintained overnight at 37 °C in 5% CO_2_.

On Day 0, cells were transferred into a 150 mL transfer bag, which was then connected using sterile welding to a TS520 Tubing Set (Miltenyi Biotec, Cat. 170-076-600) pre-installed on the CliniMACS Prodigy system. The automated isolation of CD4^+^ and CD8^+^ T-cells was performed using the CliniMACS Prodigy’s program named “T-cell transduction” (TCT 2.0) process.

The process began with the centrifugation-based removal of platelets and serum. In the next step, peripheral blood mononuclear cells (PBMCs) were incubated with magnetic beads conjugated to anti-CD4 and anti-CD8 antibodies (CliniMACS CD4 Reagent, Cat. 200-070-213; CD8 Reagent, Cat. 200-070-215; Miltenyi Biotec). Target T-cell subsets were then isolated via magnetic-activated cell sorting (MACS). A total of 1 × 10^8^ purified CD4^+^ and CD8^+^ T-cells was used for the production of CAR-T cells. The remaining selected T-cells from the Prodigy’s reapplication bag (RAB) were cryopreserved in TexMACS Medium supplemented with 5% HSA and 10% DMSO.

The preset activity matrix “Enhanced Feeding Protocol” (Miltenyi Biotec) was used without modifications for CAR-T manufacturing. T-cells in the amount of 1 × 10^8^ cells were transferred into the incubator chamber of the Prodigy machine, in TexMACS Medium supplemented with 1000–1500 U/mL IL-7 and 300–450 U/mL IL-15. T-cell activation was initiated on Day 0 via the addition of TransAct T Cell Activation Reagent (Miltenyi Biotec, Cat. 200-076-204), which promotes polyclonal activation and prepares the cells for transduction.

On Day 1, transduction was performed using 2 × 10^8^ transducing units (TUs) of an LV, either V1 (NCB.LV.CD19-CAR) or V2 (EFS.LV.CD19-CAR). On Day 3, the culture medium was replaced with fresh medium. Cells were then incubated until Day 12 according to a predefined, automated expansion program (referred to as the “activity matrix”). On Day 12, cells were automatically washed and transferred into a final product bag in TexMACS Medium. The cell concentration was manually adjusted to 1–2 × 10^7^ cells/mL, and the formulation was supplemented to final concentrations of 5% HSA and 10% DMSO. The final CAR-T cell product was then aliquoted, cryopreserved, and stored in liquid nitrogen.

### 2.7. Flow Cytometry

Antibodies for flow cytometry were obtained from Miltenyi Biotec (Table 1 Panel A). Immunophenotyping of the starting cell material was conducted using the following panel: CD3 Antibody FITC (Cat. 130-113-138), CD4 Antibody VioGreen (130-113-221), CD8 Antibody APC-Vio770 (130-113-155), CD14 Antibody APC (130-113-143), CD16 Antibody PE (130-113-393), CD19 Antibody PE-Vio770 (130-113-647), CD45 Antibody VioBlue (130-113-122), CD56 Antibody PE (130-113-312), and 7-AAD Staining Solution (130-111-568).

The CAR-T cell product was analyzed using the following antibodies (Table 1 Panel B): CD3 FITC (130-113-138), CD4 VioGreen (130-113-221), CD8 APC-Vio770 (130-113-155), CD14 APC (130-113-143), and CD45 VioBlue (130-113-122), along with the CD19 CAR Detection Reagent (Biotinylated; 130-129-550), followed by staining with Anti-Biotin Antibody PE (130-113-291).

T-cell differentiation and activation markers were assessed using the Panel C: CD3 VioBlue (130-114-519), CD4 VioGreen (130-113-230), CD8 APC-Vio770 (130-110-681), CD25 PE (130-113-286), CD45RA FITC (130-113-365), CD69 PE-Vio770 (130-112-804), and CD197 (CCR7) APC (130-120-460). T-cell activation markers were analyzed in samples collected from the incubated culture at 48 h after the addition of the activator reagent (TransAct, Miltenyi Biotec).

Subpopulations of T lymphocytes were analyzed in the final cell products as follows: naïve (Tn) cells are characterized via CD45RA^+^ CD197^+^, central memory (Tcm) cells via CD45RA^−^ CD197^+^, effector memory (Tem) cells via CD45RA^−^ CD197^−^, and terminally differentiated effector memory RA-positive (TEMRA) cells via CD45RA^+^ CD197^−^.

A single-tube staining protocol was used when possible. A sample containing ~10^6^ nucleated cells was centrifuged at 300× *g* for 10 min at RT. The cell pellet was resuspended in 100 μL of MACSQuant Running Buffer (RB; Miltenyi Biotec, Cat. 130-092-747). A cocktail containing 2 μL of each antibody from the selected panel was added to the cells. The sample was incubated for 10 min at RT in the dark. After incubation, 1 mL of RB was added, and the cells were centrifuged at 300× *g* for 5 min. The cell pellet was resuspended and used for flow cytometry analysis.

Staining for CAR transduction was performed in two stages. In the first stage, 2 μL of CD19 CAR Detection Reagent, biotin-conjugated (Miltenyi Biotec, Cat. 130-129-550), was added to a 100 μL cell suspension, followed by incubation for 10 min at RT. Subsequently, 1 mL of RB was added, and the cells were centrifuged as described above. The supernatant was discarded, and the pellet was resuspended in 100 μL of RB. The second staining step was then performed by adding a cocktail of antibodies from the selected panel along with 2 μL of anti-biotin-PE antibody (Miltenyi Biotec, Cat. 130-113-291). The cells were incubated for another 10 min at RT in the dark, washed with 1 mL of RB, and then used for flow cytometry analysis. Gating strategies are presented in Appendix A.

Samples were acquired using the MACSQuant Analyzer 10 flow cytometer. Flow cytometry data were analyzed using MACSQuantify Software v.2.13 with the installed Express Modes module (Miltenyi Biotec, Bergisch Gladbach, Germany) and FlowJo v10.8.1 (FlowJo, LLC, Ashland, OR, USA).

### 2.8. B-Cell Isolation

B cells were isolated from portions of apheresis products using density gradient centrifugation, followed by immunomagnetic cell separation. A sample of the blood product containing no more than 1 × 10^8^ nucleated cells or a volume not exceeding 15 mL was diluted with PBS supplemented with 2% human serum albumin (HSA) and 1 mM EDTA (PEB buffer) to a final volume of 30 mL. The diluted sample was layered over 15 mL of Ficoll-Paque PLUS (Cytiva (Marlborough, MA, USA), Cat. 17544203) in a 50 mL conical tube and centrifuged at 400× *g* for 30–40 min at RT without a centrifugal brake. The peripheral blood mononuclear cell (PBMC) layer was collected and transferred to a new tube. PBMCs were washed with three volumes of PEB buffer and centrifuged at 300× *g* for 10 min; this wash step was repeated once.

B cells were subsequently isolated from PBMCs using the Dynabeads Untouched Human B Cells Kit (Thermo Fisher Scientific, Cat. 11354D), following the manufacturer’s instructions.

Isolated B cells were cultured in RPMI 1640 medium (VWR, Cat. 392-0429), supplemented with 10% FBS, 1% penicillin–streptomycin, 2 mM of L-glutamine, and 50 ng/mL of B-cell-activating factor (BAFF) (Miltenyi Biotec, Cat. 130-108-987). Cells were maintained at a density of 0.5–1 × 10^6^ cells/mL and cultured for no longer than 8 days.

### 2.9. Cytokine Release Assay

Cocultures were established in 12-well plates using TexMACS Medium supplemented with IL-7 (500 U/mL) and IL-15 (150 U/mL). The cocultures consisted of anti-CD19 CAR^+^ cells, generated from patient-derived leukapheresis products, and autologous B cells isolated from the same material.

Aliquots of CAR-T cell products were thawed and maintained in short-term cultures before the experiment. The cells were distributed into 12-well plates to establish a consistent number of effector cells (1 × 10^6^ CAR^+^ cells per well) across all tested products. Target B cells were added at 1 × 10^5^ cells per well. Controls included non-transduced (NT) CD4^+^ and CD8^+^ T-cells from the reapplication bag (RAB) cocultured with B cells, and CAR-T cells alone at the same cell number as in the experimental wells to assess background cytokine production. Cocultures were incubated at 37 °C with 5% CO_2_ for 24 h. After incubation, culture media were collected, clarified, and stored at −80 °C until analysis.

Cytokine release was measured using ELISA with the Human IFN-γ ELISA Kit (Invitrogen (Carlsbad, CA, USA), Cat. KHC4021), Human TNF-α ELISA Kit (Invitrogen, Cat. KHC3011), and Human IL-2 ELISA Kit (Invitrogen, Cat. BMS221-2), according to the manufacturer’s instructions.

Measurements of cytokine release were performed in triplicate for each CAR-T cell product.

### 2.10. Cytotoxicity Assay

B cells were pelleted from short-term cultures via centrifugation at 300× *g* for 5 min and washed twice with PBS. The B cells (target cells) were then labeled with the PKH67 fluorescent dye (Sigma-Aldrich (St. Louis, MO, USA), Cat. MINI67), according to the manufacturer’s instructions. The staining reaction was quenched by adding human serum albumin (HSA) to a final concentration of 1%. The labeled cells were then washed twice with PBS containing 1% HSA and resuspended in TexMACS Medium (Miltenyi Biotec).

Cocultures were established in TexMACS Medium supplemented with reduced amounts of supportive interleukins (500 U/mL IL-7 and 150 U/mL IL-15). CAR-T cell preparations, with defined numbers of CAR^+^ cells, were seeded into wells of 12-well plates to make 1 × 10^5^, 5 × 10^5^, or 1 × 10^6^ effector cells per well. Labeled target cells were added at a fixed number of 1 × 10^5^ per well. Controls were non-transduced (NT) T-cells from the reapplication bag (1 × 10^6^ cells) cocultured with B cells (1 × 10^5^). Cocultures were incubated at 37 °C in 5% CO_2_ for 24 h. In parallel, cultures containing only PKH67-labeled target cells were maintained under the same conditions to calibrate the flow cytometer’s counting. At the end of the 24 h incubation, labeled target cells were counted using both a flow cytometer and a cell counter, the results were compared, and a coefficient was derived to correct the flow cytometer’s counts.

After incubation, cocultured cells were pelleted via centrifugation at 300× *g* for 5 min and washed with MACSQuant Running Buffer (RB). Cells were then resuspended in 100 μL of RB and analyzed using the MACSQuant Analyzer 10. The absolute number of remaining PKH67^+^ target cells was determined after excluding debris. Specific killing (%) was calculated using the following formula:Lysis of labeled cells (%) = [(PKH67^+^ cell count in control) − (PKH67^+^ cell count in coculture)]/(PKH67^+^ cell count in control) × 100.

Experiments were performed in three biological replicates, with each replicate consisting of paired coculture and control wells.

### 2.11. Statistical Analysis

Data were analyzed in GraphPad Prism v.8.4.2 (GraphPad Inc., San Diego, CA, USA) and presented on plots with median values or mean values, as described in the figure legends. Statistical significance (*p*-value, two-tailed) for differences between two groups was computed using an unpaired Mann–Whitney test. The notation for *p*-values is as follows: ns (*p* > 0.05); * (*p* ≤ 0.05); ** (*p* ≤ 0.01); and *** (*p* ≤ 0.001).

## 3. Results

### 3.1. Vector Production

Two different LVs were used in this study. One vector, NCB.LV.CD19-CAR (also referred to as V1, for brevity), encodes a CAR that replicates CTL019, the CAR in Tisagenlecleucel (Kymriah, provided by Novartis) [2]. The second vector, EFS.LV.CD19-CAR, encodes a CAR similar to the one in Axicabtagene ciloleucel (Yescarta from Kite Pharma) [21]. The first vector was packaged in-house by the authors, while the second was provided to the authors as a high-titer LV preparation from a foreign source.

Packaged vector V1 was produced via transient transfection of HEK293FT cells using a second-generation lentiviral packaging system. The method of DNA precipitation with calcium phosphate (CaPi) efficiently transfected HEK293FT cells seeded at density (100,000 cells/cm^2^) (Figure 2c,d). The high cell density in packaging cultures enabled the production of LV particles with high titers before concentration steps. The physical and biological titers of three batches of packaged vector V1 produced under identical conditions across three production rounds are shown in Figure 4. Titers in the conditioned medium exceeded 10^7^ transducing units (TU) per milliliter (Figure 4d).

Virus-containing medium was freed from cell debris via filtration through 0.45 μM filters and concentrated in two steps: first, via tangential flow filtration (TFF) using 300 kDa cartridges to reduce the volume approximately 20-fold; second, through ultracentrifugation (20,000× *g*, 2 h, 4 °C), yielding a final volume of approximately 1 mL. The centrifugation protocol used was optimized experimentally. The authors found that centrifugation at 20,000× *g* provided the best recovery of functional titers. In contrast, higher speeds resulted in decreased recovery (Figure 2e).

Figure 4a–c demonstrate the strategy to set up flow cytometer gates used to distinguish between CAR+ and CAR− cell populations. The established gates measured the percentage of CAR-transduced cells during V1 titering. Figure 4d shows functional titers in V1 vector batches obtained from three production runs. The functional titers exceeded 10^7^ TU/mL in conditioned medium, and reached 1.5 × 10^10^ TU/mL after full purification, with recovery rates of 61–87% across runs. The resulting vector preparations were suitable for clinical-scale T-cell transduction using the automated CliniMACS Prodigy system (Miltenyi Biotec).

### 3.2. CAR-T Cell Manufacturing

A timeline for the CAR-T cell manufacturing process using the CliniMACS Prodigy device, along with the programmed activity matrix (which is a list of operations performed by the cell processor), is illustrated in Figure 3.

CAR-T cell products were manufactured from 12 cryopreserved apheresis samples. Six products were generated using vector V1 (NCB.LV.CD19-CAR) and the remaining six with vector V2 (EFS.LV.CD19-CAR).

During expansion, the morphology of T-cell cultures evolved progressively. Already on the first day (Day 1) after the addition of TransAct reagent (Miltenyi Biotec), dense multicellular clusters formed, indicative of early T-cell activation (Figure 5a). By Day 5 post-activation, these clusters had dispersed, indicating that the culture transited to the proliferation and expansion phase.

T-cells demonstrated robust expansion in the system’s incubation chamber, with total cell counts rising from an initial 1 × 10^8^ to between 6.09 × 10^8^ and 3.75 × 10^9^ by Day 12 (Figure 5b,c). No statistically significant differences in expansion kinetics were observed between the V1 and V2 vector groups. Cell viability initially decreased after the start of culture across all products (Figure 5d), reaching its lowest point on Day 3 before recovering and stabilizing at consistently high levels (93–97%) through the end of the manufacturing process (Day 12).

Activation marker analysis was performed on Day 2 (48 h after TransAct addition). The majority of T-cells (75.9–93.5%) expressed the activation marker CD25, with most exhibiting a CD25^+^CD69^−^ phenotype (Figure 5e). This profile is consistent with T-cells transitioning out of early activation into expansion or early differentiation phases.

LV transduction was carried out on Day 1 at a multiplicity of infection (MOI) of 2. Transduction efficiencies were similar between vectors V1 and V2, with CAR^+^ cell percentages ranging 39.3–57.4% for V1 and 31.6–51.4% for V2 (Figure 5f).

The CliniMACS Prodigy process with either vector generated a sufficient number of CAR^+^ cells for clinical application (Figure 5g). Even the product with the lowest CAR^+^ cell count in this study contained 3.36 × 10^8^ CAR^+^ cells, sufficient to treat a 100 kg patient at the standard clinical dose of 1 × 10^6^ cells/kg.

### 3.3. Analysis of Cellular Subpopulations in CAR-T Products

All manufactured CAR-T cell products were highly enriched for T-cells, with CD3^+^ populations constituting >99% of cells and CD3^−^ fractions representing <1%. Flow cytometry analysis using antibody panel B (Table 1) confirmed the absence of detectable inhibitory cell types, such as monocytes. Diagrams from flow cytometry experiments and cellular composition summary are provided in Appendix A.

In most manufactured cell products, CD4^+^ T-cells outnumbered CD8^+^ T-cells. The CD4/CD8 ratio in CAR^+^ cells showed no significant difference between vector V1 (median = 1.87) and vector V2 (median = 2.24), and the ratio was not significantly different from that observed in CAR^−^ cells across all products (Figure 6a,b).

CD4^+^ CAR^+^ cells were predominantly composed of central memory T-cells (Tcm; mean: 56.9% ± 13.4% across 12 products) and effector memory T-cells (Tem; 42.4% ± 13.1% across 12 products). Naïve T-cells (Tn) were virtually absent (<0.1%), while the proportion of terminally differentiated effector memory cells re-expressing CD45RA (Temra) was minimal (<0.9%) (Figure 6c,d). No statistically significant differences in the distribution of these subsets were observed between the V1 and V2 vectors for CD4^+^ CAR^+^ cells.

Representative flow cytometry dot plots illustrating the CD45RA/CCR7 gates setup using leukapheresis material from a healthy donor, as well as the analysis of CD8^+^ CAR-T cell subsets, are shown in Figure 6e–g. The analysis showed that CD8^+^ CAR^+^ cells consisted primarily of Tcm (44.1% ± 16.6%) and Tem (55.4% ± 16.8%) subsets across all products. As with the CD4^+^ CAR^+^ population, Tn cells were virtually undetectable among CD8^+^ CAR^+^ cells by flow cytometry, and Temra cells were present at very low levels (maximum 0.8%) (Figure 6h,i). Notably, CD8^+^CAR^+^ cells exhibited statistically significant differences in dominant subsets depending on the vector used. Products generated with vector V1 showed significantly higher proportions of Tcm cells compared to V2 (mean ± SD: 58.3% ± 7.8 vs. 29.9% ± 7.6; *p* = 0.0022). Conversely, Tem cell frequencies were markedly elevated in V2-derived products (69.4% ± 8.3 vs. 41.5% ± 7.8; *p* = 0.0022) (Figure 6h,i).

### 3.4. Functional Analysis of CAR-T Cells

The functional activity of two CAR^+^ cell variants, generated with vectors V1 and V2, was evaluated by co-culturing the effector cells (E) with CD19^+^ target cells (T) at an effector-to-target (E:T) ratio of 10:1. Production of key Th1 cytokines (IFN-γ, TNF-α, and IL-2) was measured to assess CAR-T cell activation. All three cytokines were significantly elevated in cocultures after 24 h (elevation was highest in IFN-γ, followed by TNF-α, then IL-2), confirming antigen-specific activation of CAR^+^ T-cells (Figure 7a). In contrast, minimal cytokine production was detected in control wells containing either non-transduced (NT) T-cells from the CliniMACS Prodigy reapplication bag cocultured with B cells, or CAR^+^ cells cultured without target cells (baseline production, BL).

Notably, CAR^+^ cells generated using the V2 vector exhibited, on average, higher production of the three cytokines tested, compared to those generated with V1 (IFN-γ: 1.7-fold, *p* = 0.015; TNF-α: 2.7-fold, *p* = 0.002; IL-2: 4.4-fold, *p* = 0.002), suggesting enhanced in vitro functional potency of V2-derived CAR^+^ cells (Figure 7a).

The cytotoxic activity of CAR^+^ cells was evaluated by quantifying lysis of fluorescent dye PKH67-labeled CD19^+^ target cells in cocultures at effector-to-target (E:T) ratios of 10:1 and 5:1. Despite differences in cytokine production, V1- and V2-derived CAR^+^ cells demonstrated comparable cytotoxicity. The V2 group demonstrated a trend toward enhanced target cell lysis (Figure 7b,c), although the intergroup difference did not reach statistical significance. Minimal lysis of labeled cells was observed in control cocultures with non-transduced (NT) T-cells.

In summary, in vitro tests confirmed that CAR^+^ cells generated with both vectors were functionally competent, with V2 demonstrating superiority in cytokine production.

## 4. Discussion

This work describes the first complete CAR-T manufacturing pipeline in Central Asia, and it represents a leap forward in making this life-saving therapy accessible for patients with relapse or refractory hematologic malignancies in our geographic region. Kazakhstan is a country with 20 million people and a developing economy. Each year, approximately 1300 new cases of leukemia, lymphoma, and multiple myeloma are diagnosed in the country, and about 300 patients annually develop a relapse or refractory disease, with no more effective treatment options available to date domestically.

While CAR-T cell therapy has revolutionized the treatment of B-cell malignancies worldwide since its first regulatory approval in 2017 [1,2], it has not yet been introduced in our part of Eurasia. This was primarily due to the lack of manufacturing capacity for clinical-grade CAR-T cell products and the absence of trained medical personnel with hands-on experience in clinical application. Only a handful of patients from Kazakhstan have received CAR-T therapy through self-funded medical tourism, despite prohibitively high costs.

Beyond the high cost of industrial CAR-T like Kymriah and Yescarta, additional barriers hinder their adoption in Kazakhstan and across all Central Eurasia. To date, Kazakhstan’s national regulatory authority has not approved any foreign CAR-T therapies for clinical use. One more challenge is the complex logistics of cellular material between domestic clinical sites and international manufacturing facilities. Geographical remoteness and cross-border regulatory constraints create substantial operational barriers. Another critical barrier is the absence of CAR-T-trained physicians in Kazakhstan, particularly specialists capable of preparing patients for therapy, administering CAR-T treatments, and managing post-treatment complications.

Two models of CAR-T cell production currently coexist worldwide. The first, often termed the “industrial format,” relies on large-scale manufacturing in centralized biotech facilities that distribute standardized CAR-T products to multiple treatment centers. These therapies require full regulatory approval, such as from the FDA, EMA, etc., before clinical use [24,25].

In contrast, the “academic model” is a decentralized approach, through which CAR-T cells are produced locally (typically in hospital-affiliated labs) and used exclusively at that center. In many cases, such therapies have not undergone full clinical trials and are administered under special regulatory provisions for unlicensed medicines, known as “hospital exemption” rules [26]. Point-of-care production and clinical application under hospital exemption frameworks reduce CAR-T therapy costs by 60–80% compared to commercially approved products. Published estimates from Western Europe indicate academic CAR-T production costs approximately €80,000 per dose, versus €375,000 for industry-approved equivalents [27,28].

Such a significant cost reduction is critical in resource-limited settings like ours, where financial constraints remain the foremost barrier to accessing CAR-T cell therapy. Our study marks the establishment of Kazakhstan’s first academic CAR-T cell manufacturing platform, addressing two critical milestones: the development of domestic lentiviral vector production capacity and the implementation of a GMP-compatible automated CAR-T manufacturing process using the CliniMACS Prodigy system.

Lentiviral (LV) and gamma-retroviral vectors (gRVs) are foundational to the CAR-T field: they can efficiently transduce hematopoietic cells, including primary T-cells, and stably integrate transgenes into the host genome for durable therapeutic expression. Indeed, all currently approved CAR-T therapies utilize only LV or gRV for gene delivery, with no alternative methods adopted in clinical manufacturing to date [29,30,31].

Pre-packaged clinical-grade lentiviral vectors represent a major cost driver in CAR-T manufacturing, accounting for 10–30% of total production expenses [32,33,34,35]. In-house LV production is critical for cost reduction in academic CAR-T programs, avoiding expensive externally sourced vectors. Compounding the issue of vector accessibility, intellectual property protections on clinically validated viral vectors used by industrial CAR-T manufacturers restrict access for third-party producers and pose a significant barrier to the broader dissemination of CAR-T therapies.

We present an optimized in-house LV production protocol using calcium phosphate transfection of high-density HEK293FT cultures (100,000 cells/cm^2^), yielding consistent titers of 1.5 × 10^7^–4.1 × 10^7^ TU/mL in unconcentrated medium. A two-step concentration yields 1.3–1.5 × 10^10^ TU/mL, compatible with clinical-scale CAR-T production. In this study, we evaluated two LVs that differ only in their cloned CAR gene while sharing identical backbone structures. The CAR gene in vector V1 replicates the architecture of CTL019, the precursor to Novartis’ Kymriah, while vector V2 incorporates the published design of Kite/Gilead’s Yescarta.

To compare vector performance during CAR-T cell production, we manufactured twelve CAR-T products (six per construct) using the closed, automated CliniMACS Prodigy platform. This GMP-compliant system ensured standardized processing conditions across all manufacturing runs. Both lentiviral vectors demonstrated comparable performance in transduction efficiency, CAR^+^ T-cell expansion, and viability, despite inherent donor-to-donor variability. These findings confirm that the lentiviral vector packaging and purification strategy described herein reliably yields vectors suitable for clinical-scale manufacturing.

An immunophenotypic analysis revealed that CAR^+^ cells from both vectors were predominantly CD4^+^ T-cells, and that both CD4^+^ and CD8^+^ subsets showed a central and effector memory (Tcm/Tem) phenotype, with minimal naïve (Tn) or terminally differentiated populations. This memory-skewed profile likely results from the CliniMACS Prodigy platform’s cytokine milieu, which incorporates IL-7 and IL-15, both known to preferentially support memory T-cell expansion. The near-complete absence of naïve T-cells was indicative of a shift toward Tcm. We observed significant vector-dependent differences in CD8^+^ T-cell memory phenotypes: V1 induced a substantially higher proportion of CD8^+^ central memory T-cells (Tcm) than V2 (58.3% vs. 29.9%; *p* < 0.01), phenotypically mirroring the superior persistence associated with 4-1BB–containing CARs such as Kymriah [36]. Conversely, the V2 group showed elevated frequencies of CD8^+^ effector memory T-cells (Tem; 69.4% vs. 41.5%; *p* < 0.01), consistent with the effector-dominant phenotype typically seen with CD28-costimulated CARs like Yescarta [36,37].

While our study demonstrates functional differences between CAR-T cells generated with different vectors, it is important to note that the CARs in V1 and V2 differ not only in their costimulatory domains but also in the hinge and transmembrane regions (Figure 1). These structural differences may influence CAR+ cell effector functions in addition to the costimulatory domains. For example, the CD28-derived hinge/transmembrane region in the Yescarta-like construct (V2) may promote stronger immunological synapse formation or alter CAR surface expression compared to the CD8-derived regions in the Kymriah-like construct (V1) [36,38,39,40]. Thus, the observed differences in cytokine production (e.g., higher Th1 cytokines with V2) and subset distribution (e.g., Tem skewing in CD8^+^ V2 CAR-T cells) could reflect combined effects of costimulation and structural disparities. Future studies will help dissect the individual contributions of each structural region to further optimize the CAR design.

All Prodigy-manufactured anti-CD19 CAR-T products showed antigen-specific function via CD19-dependent cytokine release. However, CD28-based CAR^+^ cells (V2) showed significantly higher Th1 cytokine production (IFN-γ, TNF-α, IL-2) versus 4-1BB-based CAR^+^ cells (V1), aligning with CD28’s potent T-cell activation role [36,38].

CAR-T cells generated with both vectors showed comparable cytotoxicity, though V2 (CD28) CAR^+^ cells trended toward greater killing capacity.

CliniMACS Prodigy instruments have become a de facto industry standard for academic-scale CAR-T cell production for clinical usage, across the world. Published examples of the successful deployment include Brazil [13], Thailand [14], Spain [37], Mexico [41], and Russia [42]. Our results align with key metrics from the cited studies, e.g., Salazar-Riojas et al. (2025) reported a median transduction efficiency of 44.7% and a viability of 97% in final products [41], closely matching our collected data. The cited article presents an estimated production cost of ~$32,000 per dose, underscoring the economic advantage of academic CAR-T manufacturing, which is a 10–20-fold cost reduction compared to commercially available CAR-T therapies. Our immunophenotypic analysis revealed a predominance of memory T-cell subsets (Tcm/Tem) in the final CAR-T products, consistent with a recent report using the Prodigy system [37].

Our findings demonstrate the feasibility of establishing an academic CAR-T manufacturing platform in a developing country with no prior experience in cell and gene therapy (CGT) or CAR-T. The use of Miltenyi Biotec’s technologies enables the production of cell therapy products that are potentially suitable for clinical use. Thus, Miltenyi Biotec’s solutions help rapidly bridge the technological gap between countries where CAR-T therapy has been implemented and those with no prior access to this advanced technology.

Drawing on the experience in Kazakhstan, we conclude that upscaling the production and cutting costs to levels acceptable to the local health insurance system requires addressing several tasks. In-house vector production significantly reduces initial CAR-T manufacturing costs, especially during trial runs. Addressing the production of clinical-grade CAR-T cells, which requires cleanroom conditions, skilled personnel, and validated protocols, can reach manufacturing readiness in a relatively short time when using established technologies such as automated cell processors like the CliniMACS Prodigy.

For biotech labs in academic settings without access to costly Class A/B cleanrooms, fully closed systems like CliniMACS Prodigy enable cell manufacturing in ISO 7 (Class C) environments. Only a few steps, such as cytokine addition, require ISO 5 (Class A) conditions and can be performed in a laminar flow cabinet. Automated platforms offer the advantage of standardized protocols, ensuring consistent product quality even in settings with limited technical proficiency. Successful CAR-T adoption in the untapped markets of developing countries requires not only technology acquisition but also supportive regulatory frameworks. In Kazakhstan, the national regulator allows academically produced advanced therapy medicinal products (ATMPs) to be used under hospital exemption regulations.

A critical unresolved challenge in our country remains the lack of medical personnel trained in CAR-T therapy. To address this gap, we strongly recommend prioritizing hands-on clinical training early during the project. Such training should cover all essential aspects of CAR-T therapy, including patient safety protocols and toxicity management. This prioritization stems from the lack of dedicated international or private CAR-T training centers for physicians worldwide. Thus, securing training opportunities in CAR-T for Kazakhstani clinicians has required extensive negotiations with foreign medical institutions, a process that remains ongoing. Furthermore, the lack of internationally recognized, standardized CAR-T training programs for clinicians makes it difficult for national regulators to objectively evaluate clinical teams’ competency in CAR-T therapy administration. In our country, the hospital exemption framework facilitates and accelerates clinical translation of CAR-T products. Based on our project experience, we recommend initiating early and sustained engagement with national regulatory authorities, because in a country without prior use of cell and gene therapy (CGT), the regulator may be unfamiliar with CAR-T technology’s unique requirements. These include, as examples, challenges in standardizing CAR-T products, the range of safety tests required, the scope of preclinical testing in animals, and related models. Regulators from developing countries will also benefit from training in CGT/CAR-T (in parallel with clinicians), to ensure the accurate evaluation of these products.

## 5. Conclusions

This article presents a successfully implemented approach for establishing CAR-T cell therapy production in an academic laboratory setting within a developing country with no prior experience. Leveraging established technologies, a highly automated cell processor, and locally produced vectors can significantly accelerate the path to manufacturing readiness.

## Figures and Tables

**Figure 1 biomolecules-15-01166-f001:**
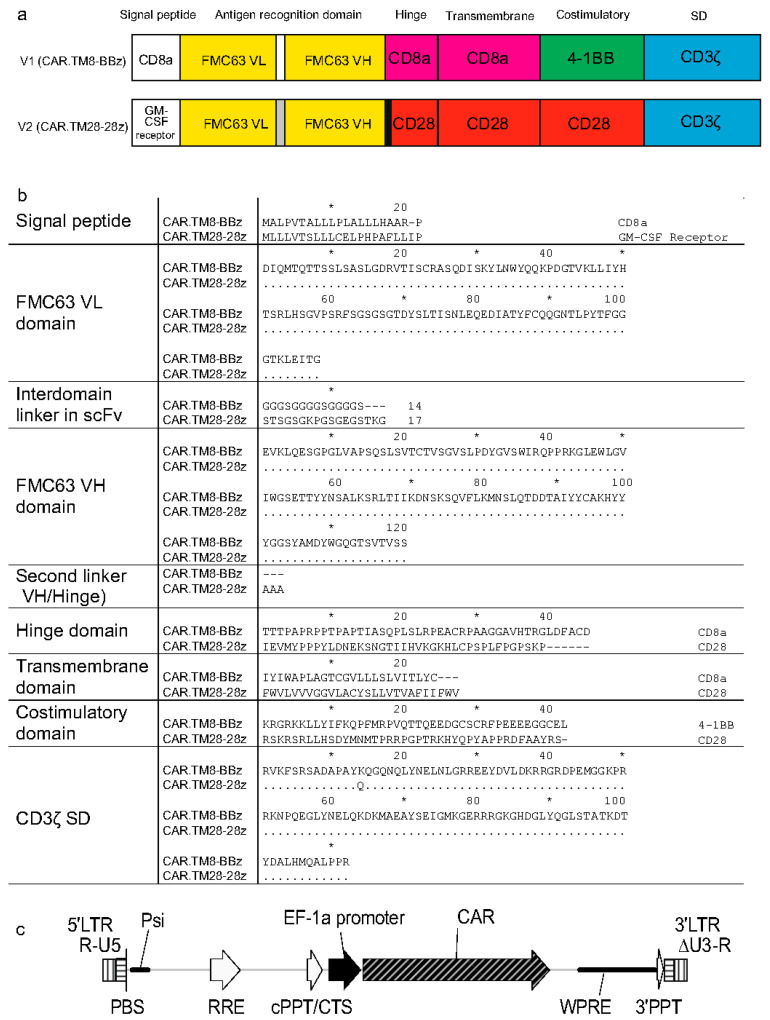
Comparison of two CAR constructs and a genetic map for lentiviral vectors in this work. (**a**) Domain organization of CAR.TM8-BBz (encoded by vector V1) and CAR.TM28-28z (encoded by vector V2). Domains that are identical in the two CARs are color-coded consistently. (**b**) Sequence alignment of domains and linkers, with amino acid residues numbered at 10-residue intervals. Identical residues are denoted with asterisks (*) and gaps with dashes (-). Domains are annotated with their respective source proteins. (**c**): Vectors V1 and V2, used to deliver the respective CAR genes, share identical genetic maps and are both second-generation lentiviral vectors. Gene abbreviations are listed in the abbreviations section.

**Figure 2 biomolecules-15-01166-f002:**
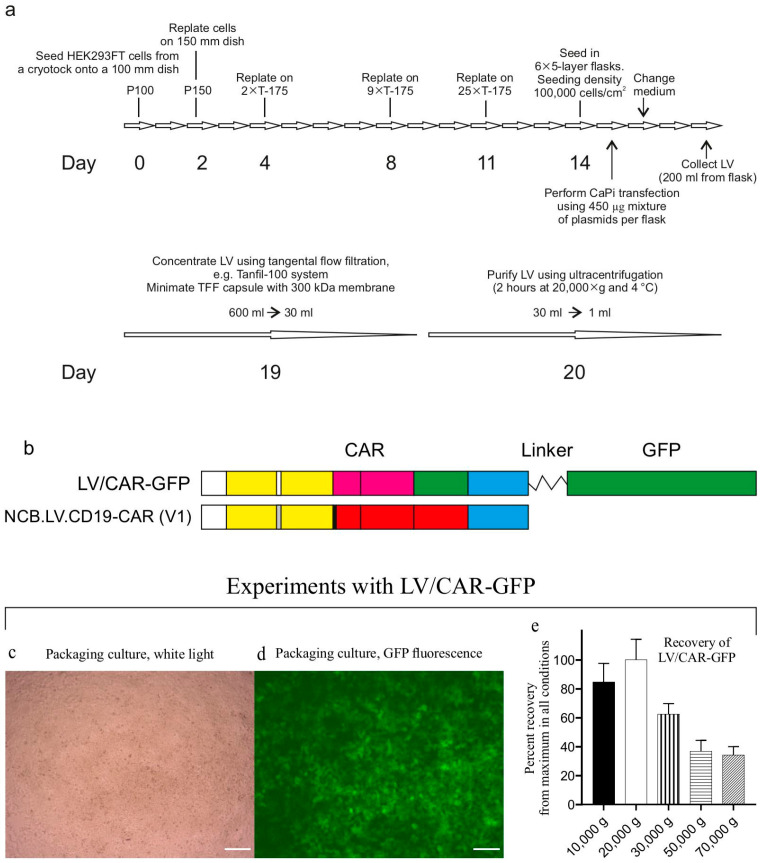
Workflow for lentiviral vector (LV) production and results with the control vector LV/CAR-GFP. (**a**) Schematic of the vector production process. Packaging cells were expanded in progressively larger culture vessels (100 mm dishes, T-175 flasks, 5-layer stacks) before calcium phosphate transfection with the LV packaging system (transfer vector, Gag/Pol helper, and VSV-G envelope plasmid). Conditioned medium was harvested 48 h post-transfection, clarified, and filtered. Viral particles from ~600 mL supernatant underwent concentration through tangential flow filtration (TFF), followed by ultracentrifugation, yielding 1 mL of high-titer preparation. These procedures provided a sufficient viral stock for multiple CAR-T cell manufacturing runs. (**b**) Comparison of the transgene in the control vector LV/CAR-GFP and the experimental vector NCB.LV.CD19-CAR. In LV/CAR-GFP, the CAR gene is fused to GFP via an uncleavable linker. In similar experiments, amounts of GFP-fluorescent cells served as an indicator of transfection efficiency. (**c**–**e**) Results of preliminary experiments with the CAR-GFP-expressing vector. The photographs on panels (**c**) and (**d**) show typical results from transfection for LV packaging. Calcium phosphate precipitation method allows the efficient transfection of high-density cultures (100,000 cells/cm^2^). The photographs were taken at 48 h post-transfection. Magnification = 50×. Scale bar = 100 µm. (**e**) Determining the optimal acceleration for centrifugal concentration of LV particles. The average functional titers are shown as means ± SDs, expressed as a percentage of the maximum titer across all experiments. The titer obtained at 20,000× *g* was set to 100%.

**Figure 3 biomolecules-15-01166-f003:**
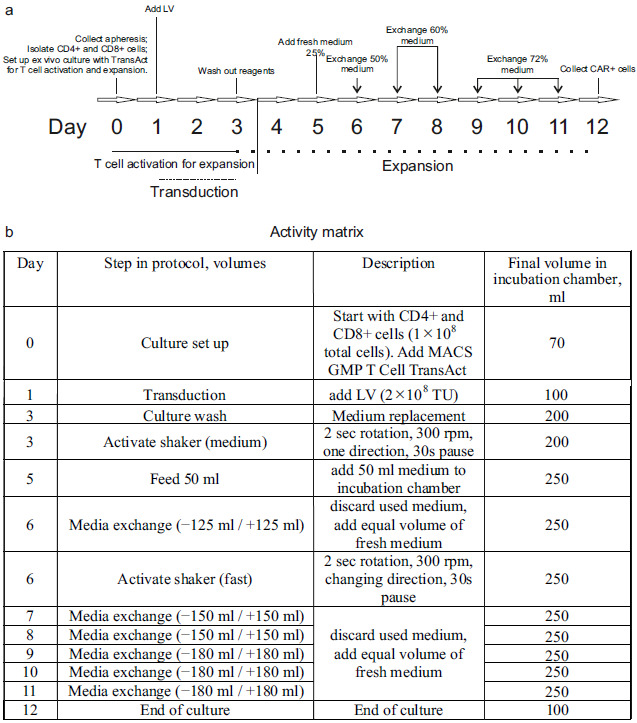
Miltenyi Biotec technology for automated clinical-grade CAR-T cell manufacturing using the CliniMACS Prodigy platform. (**a**) CAR-T Cell manufacturing timeline. Key production steps: (i) T-cell isolation and activation (Day 0)—CD4^+^/CD8^+^ T-cells are immunomagnetically selected from leukapheresis product and activated using GMP-compliant T-Cell TransAct reagent; (ii) lentiviral transduction (Day 1); and (iii) automated expansion culture (Days 3–12) with continuous monitoring. (**b**) Instrument control interface showing the activity matrix that directs the automated manufacturing process. The closed-system design ensures minimal manual handling while maintaining GMP compliance throughout cultivation, media exchange, and harvest procedures.

**Figure 4 biomolecules-15-01166-f004:**
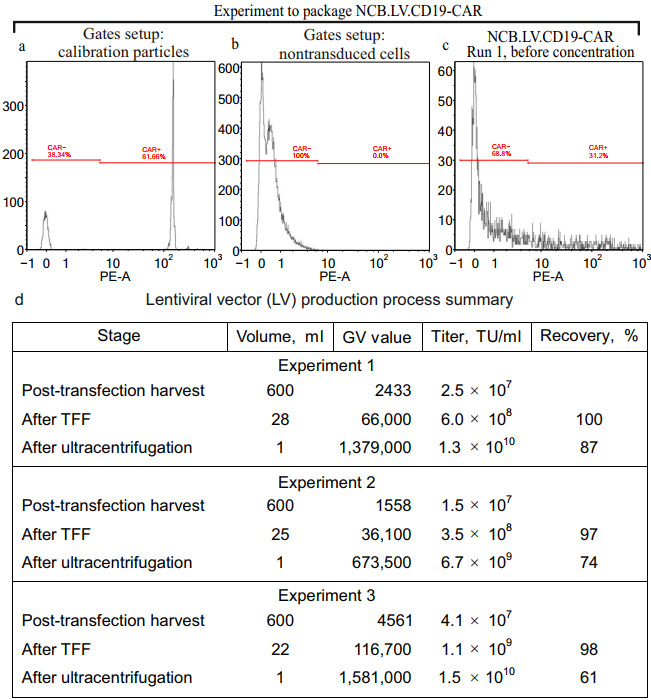
Titer measurement of packaged vector preparations. (**a**) Fluorescent calibration particles (from Miltenyi Biotec) were used to establish flow cytometry gates for CAR-positive (CAR+) and CAR-negative (CAR−) populations. (**b**) Naïve HEK293 cells were analyzed to validate flow cytometry gating. No CAR+ cells were detected, confirming gate specificity. (**c**) Representative biological titer analysis of V1 vector (Batch Run 1). Sample shows cells transduced with unconcentrated culture medium (diluted 1:200). (**d**) Summary of three production batches. Data presented: volume of virus-containing material, physical titer (Lenti-X GoStix GV units; 1 GV = 1 ng/mL p24), functional titers (transducing units, TU/mL), recovery rates (%).

**Figure 5 biomolecules-15-01166-f005:**
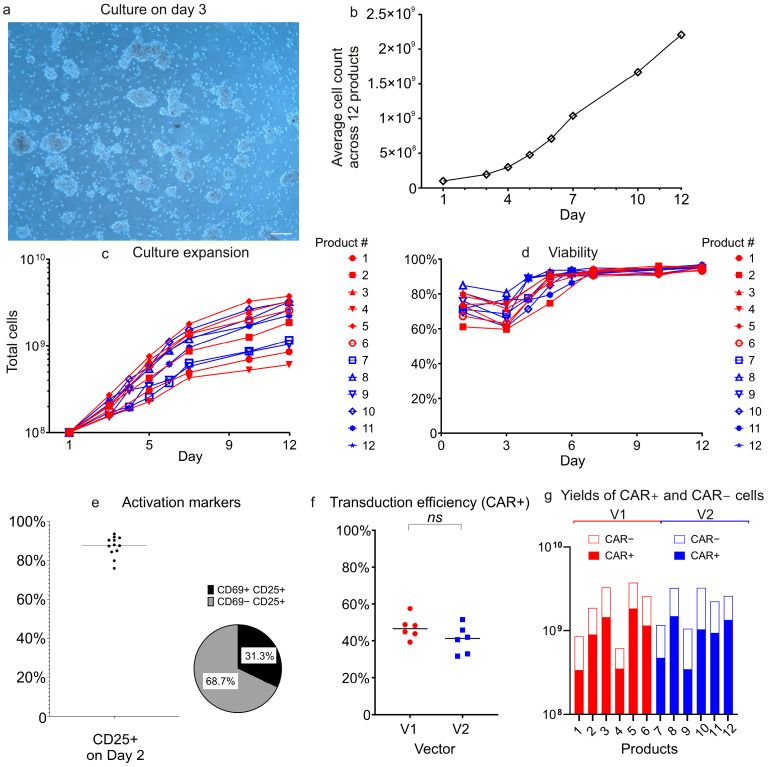
CAR-T cell manufacturing using vectors V1 and V2. (**a**) A microscopic image of cells in the CliniMACS Prodigy incubation chamber on Day 3. Visible cell clustering is indicative of T-cell activation. Magnification = 10×. Scale bar = 100 µm. (**b**) Time-course analysis of mean cell counts from twelve independently manufactured CAR-T cell products shows robust expansion from 1 × 10^8^ to 2.2 × 10^9^ (average) cells during 12 days of in vitro expansion. There was no statistically significant difference between vectors V1 and V2. (**c**) Cell growth kinetics in individual cultures (n = 12; V1 in red, V2 in blue). (**d**) Cell viability increases from Day 3 to Day 6 and remains >93% through Day 12; (**e**) T-cell activation markers: >75% of cells express CD25 by Day 2; most are CD25^+^CD69^−^, indicating ongoing expansion. (**f**) Transduction efficiency (% CAR^+^ cells) is similar for V1 and V2. (**g**) Yield of CAR^+^ cells is sufficient for therapeutic use in all products. (**c**,**d**) show individual sample counts over time; (**e**,**f**) present dot plots with medians. The pie chart in panel (**e**) displays CD69 expression among CD25^+^ cells; (**g**) shows the absolute numbers of CAR^+^ cells (filled bars) and untransduced cells (open bars) in the same products, by the vector used; *ns* = not significant.

**Figure 6 biomolecules-15-01166-f006:**
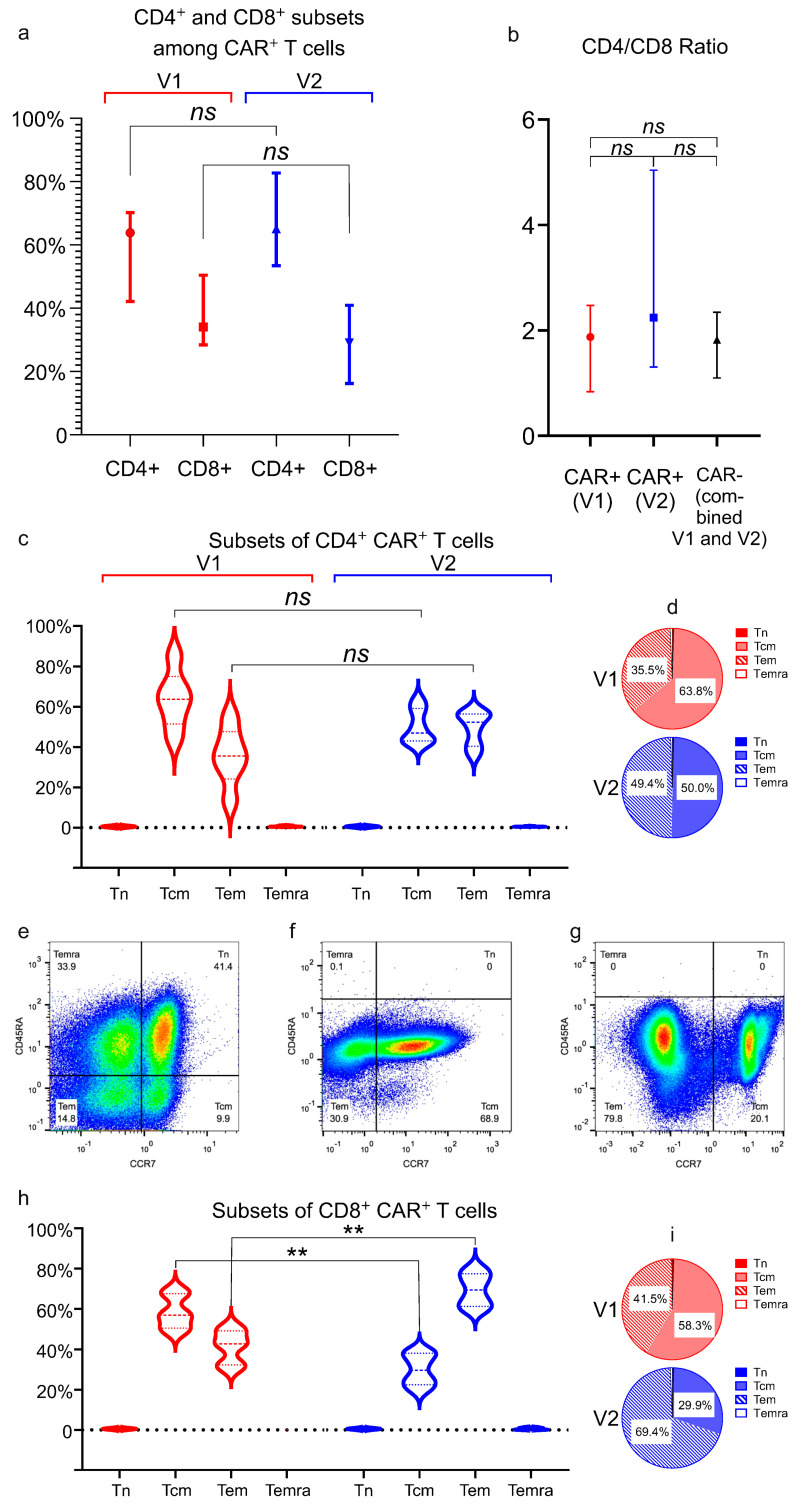
Analysis of CD4+ and CD8+ subsets among CAR+ T cells. (**a**) Distribution of CD4^+^ and CD8^+^ cells in CAR^+^ T-cell populations. No significant differences between V1 and V2 groups were found; (**b**) CD4/CD8 ratio in CAR^+^ T-cells (V1: n = 6; V2: n = 6) compared to untransduced (CAR^−^) T-cells (n = 12); (**c**) subset analysis of CD4^+^ CAR^+^ T-cells, proportions of naïve (Tn), central memory (Tcm), effector memory (Tem), and terminally differentiated effector (Temra) cells in different CAR-T products are shown as violin plots. There are no significant differences between V1 and V2 groups of CD4^+^CAR^+^ T-cells; (**d**) Pie charts depict the distribution among CD4^+^CAR^+^ T-cells with immunophenotypes of naïve (Tn), central memory (Tcm), effector memory (Tem), and terminally differentiated effector (Temra) cells; (**e**) Dot plot depicts cross-hair gating used to separate T-cell subsets (Temra, Tn, Tem, Tcm) in leukapheresis material from a healthy donor, based on CD45RA and CCR7 expression; (**f**) Representative data for a CAR-T cell product generated with the V1 vector. The cell population gated on CD4^+^ cells is displayed, showing the distribution of T-cell subsets (CD45RA vs. CCR7); (**g**) Representative data for a CAR-T cell product generated with the V2 vector. The cell population gated on CD8^+^ cells is displayed, showing the distribution of T-cell subsets (CD45RA vs. CCR7); (**h**) Analysis of CD8^+^CAR^+^ T-cells revealed vector-dependent subset distribution, with V1 favoring Tcm and V2 favoring Tem. Significant differences were observed between vectors for these subsets (*p* < 0.01); (**i**) Pie charts depict the distribution among CD8^+^CAR^+^ T-cells of Tn, Tcm, Tem, Temra, stratified by vector. Panels (**a**,**b**) display medians with 95% confidence intervals; (**c**,**h**) show violin plots with medians and interquartile ranges; (**d**,**i**) show mean shares from total; The red color represents group V1, the blue color represents group V2, and the black line on panel (**b**) shows untransduced cells in the combined groups V1 and V2; ns = not significant, ** *p* ≤ 0.01.

**Figure 7 biomolecules-15-01166-f007:**
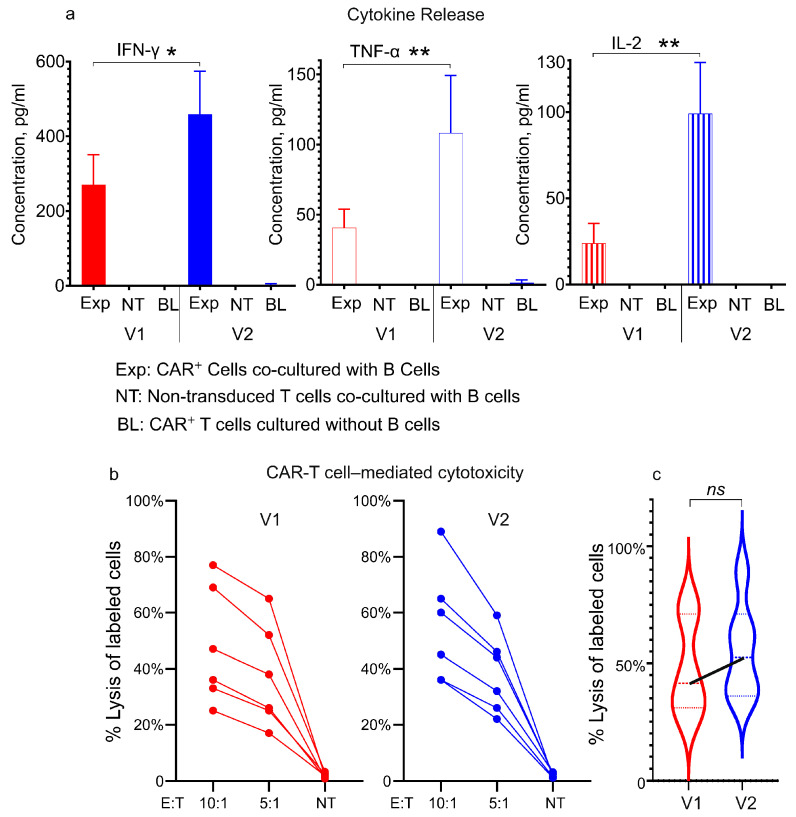
Functional characterization of CAR-T cells via cytokine secretion and cytotoxicity assays. (**a**) Key Th1 cytokine release profile. CAR^+^ T-cells were cocultured with CD19^+^ targets (B cells) (Exp) and compared to controls: non-transduced T-cells (NT) cultured with targets, and CAR^+^ T-cells without targets (baseline, BL). Coculture of CAR^+^ effectors with targets induced significant production of IFN-γ, TNF-α, and IL-2, indicating antigen-specific activation. Data shown as means ± SDs (V1: n = 6; V2: n = 6); (**b**) cytotoxic activity of CAR^+^ T-cells was assessed by measuring lysis of labeled target cells at effector-to-target (E:T) ratios of 10:1 and 5:1. Control cocultures contained non-transduced (NT) T-cells with labeled targets. Both V1- and V2-derived CAR^+^ cells exhibited dose-dependent target cell lysis (increasing with higher E:T ratios). Each data point represents the mean of triplicate measurements per sample; (**c**) violin plots with connected medians show a trend toward enhanced cytotoxicity in V2- vs. V1-derived CAR^+^ T-cells; ns = not significant, * *p* ≤ 0.05, ** *p* ≤ 0.01.

**Table 1 biomolecules-15-01166-t001:** Antibody panels used in the study.

Antibody	Panel A	Panel B	Panel C
	Cell Populations in Starting Cell Material	CAR Transduction	Immunophenotyping T-Cell Subsets
CD3 Antibody FITC (130-113-138)	√	√	
CD3 Antibody VioBlue (130-114-519)			√
CD4 Antibody VioGreen (130-113-221)	√	√	
CD4 Antibody VioGreen™ (130-113-230)			√
CD8 Antibody APC-Vio770 (130-113-155)	√	√	
CD8 Antibody APC-Vio 770 (130-110-681)			√
CD14 Antibody APC (130-113-143)	√	√	
CD16 Antibody PE (130-113-393)	√		
CD19 Antibody PE-Vio770 (130-113-647)	√		
CD25 Antibody PE (130-113-286)			√
CD45 Antibody VioBlue (130-113-122)	√	√	
CD45RA Antibody FITC (130-113-365)			√
CD56 Antibody PE (130-113-312)	√		
CD69 Antibody PE-Vio 770 (130-112-804)			√
CD197 (CCR7) Antibody APC (130-120-460)			√
7-AAD (130-111-568)	√	√	
CAR Detection Reagent, Biotin (130-129-550)		√	
Biotin Antibody PE (130-113-291)		√	

## Data Availability

The data is contained within the article or Appendix A. The dataset is available upon request from the authors.

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
