# Peer review of "Introducing CAR-T Therapy in Kazakhstan: Establishing Academic-Scale Lentiviral Vector and CAR-T Cell Production"

_biomolecules, 2025, doi:10.3390/biom15081166_

Round 1
Reviewer 1 Report
Comments and Suggestions for Authors
Summary: CAR-T cell therapies are curative for many hematological malignancies, but are challenging to manufacture especially in under-resourced countries. The authors establish Kazakhstan’s first CAR-T production pipeline including both viral vector manufacturing and CD19 CAR-T manufacturing on the CliniMACS Prodigy instrument. While the novelty of the data presented is limited, the manuscript is valuable for establishing the first successful CAR-T pipeline in Kazakhstan, as well as delivering detailed methods explaining their viral vector manufacturing and CAR-T expansion methods. In addition to the excellent level of detail, the authors are also commended for presenting a very robust data set that includes 12 donors split across 2 manufacturing groups (vector 1 vs. vector 2).
Major comments:
- Figure 5b – It is recommended to add an additional panel to the Figure which displays the average total cell count on day 12, and perhaps also an earlier timepoint(s), where the reader can easily read out the average expansion.
- Figure S6 – there appear to be some potential issues in the gating. First, could the authors please explain what is the purpose of the top right graph “CAR expression” on the graph of 7-AAD vs. CAR-PE? Second, and more importantly, the CD4 and CD8 gates appear to show issues with compensation as the CD8+ population is also CD4-positive. This would indicate potential problems in the staining panel that could lead to data misinterpretation.
- Figure 6 – Please add representative flow plot(s) of the CD45RA+CCR7+ subset gating either in Figure 6 or in an additional supplementary Figure. Since there is a significant difference between V1 and V2 within the CD8 compartment, it is recommended to show representative flow plots of this phenomenon.
- Discussion – Others have published CliniMACS Prodigy-based manufacturing of CAR-T cells, could the authors compare their method / results with other published results?
Minor comments:
- Section 2.5 – these were not healthy donors, but rather unclaimed/unused products collected for treatments. While the authors are unable to access specific patient information, could they perhaps share what types of conditions the center would collect these products for? For example, are these potentially leukemia patients?
- Line 260 – CAR-T expansion. The authors mention a rest step after thaw (day -1 to day 0), but interestingly add IL7 and IL15 cytokines. Adding cytokines (signal 3) without CD3/28 (signals 1 and 2) can promote T cell anergy. In future adaptations of the expansion, it is recommended to withhold cytokines during rest/recovery. In addition, could the authors please add the vessel used for recovery step (i.e., flask, bag, etc).
- Figure 5a – While the image on Day 3 is helpful, the images presented on Day 5 and Day 12 are not clear, and Day 12 is very dark. This could be due to mixing being “on” and thus there are no T cells present in the image plane, or it could be improper focusing of the microscope. It is recommended to replace these images or delete. In addition the text states “darkly stained aggregates”, but, it is important that there is no stain present and thus it is recommended that “darkly stained” be removed from text.
- Figure 7 – it would be helpful to graph IFN-g, TNF-a, and IL-2 in separate graphs in order to adjust the axes, so that the reader can more easily visualize the differences between V1 and V2.
- Section 3.4, Functional Analysis – the authors isolated B cells from the apheresis products instead of using a CD19+ cell line. This could present potential issues due to donor variability of B cell quality, especially for example comparing V1 vs V2 groups. Could the authors explain the rationale? Please also state in the methods if B cells were isolated from all 12 products, or from a subset of products.
Author Response
Reviewer 1 wrote: “CAR-T cell therapies are curative for many hematological malignancies… The authors establish Kazakhstan’s first CAR-T production pipeline… While the novelty of the data presented is limited, the manuscript is valuable for establishing the first successful CAR-T pipeline in Kazakhstan,.. In addition to the excellent level of detail, the authors are also commended for presenting a very robust data set…”
Response: We sincerely appreciate Reviewer 1’s thoughtful recognition of our work on introducing CAR-T therapy, the efficient yet technologically complex treatment, to Kazakhstan, our country with a developing economy. While CAR-T has never been used in our country and Central Asia, there is urgent demand from patients and physicians. This demand motivates our efforts, and we hope that the technology we have developed and described in this article will enable access to CAR-T in the country.
Reviewer 1 wrote: “Figure 5b – It is recommended to add an additional panel to the Figure which displays the average total cell count on day 12, and perhaps also an earlier timepoint(s), where the reader can easily read out the average expansion”
Response: This has been done. Figure 5 in the revised manuscript now includes panel 5b, which displays the average total cell count from day 1 to day 12 (averaged across 12 cell products). This addition provides readers with immediate insight into the average culture expansion dynamics.
Reviewer 1 wrote: “Could the authors please explain what is the purpose of the top right graph “CAR expression” on the graph of 7-AAD vs. CAR-PE?”
Response: We appreciate the reviewer's thoughtful analysis of the flow cytometry panel 7-AAD/CAR-PE (upper right corner) in Supplementary Material Figure S7 (S7 in the revised version). Why this panel has been included in the original figure: because it is a part of a pre-set of panels which are generated by our flow cytometer (Miltenyi Biotec’s MACS Quant) during automated analysisi of CAR-T cell products. More specifically, we use Miltenyi Biotec’s technology which includes the Prodigy (cell processor) and MACS Quant (cytometer) as well as cytometer-installed software. The module “CART_Transduction” in the MACS Quantify software is used to analyze CAR-T cells and results were presented in the figure. What is important is that the "CAR Expression" panel (7-AAD/CAR-PE) is technically redundant, because:
- The actual CAR expression data used by the software for statistics originates from a different panel (CD3/CAR) which is also present among the panels.
- The CAR+ cell populations for subset analysis are gated from the CD3/CAR panel, not the above mentioned 7-AAD/CAR-PE.
For these reasons, and to prevent potential reader confusion, we have removed the redundant "CAR Expression" panel from the updated Supplementary Material. All essential CAR expression data are still present in the CD3/CAR panel.
This revision of the Supplementary Material includes a modified figure.
Reviewer 1 wrote: “Figure S6 – there appear to be some potential issues in the gating. Second, and more importantly, the CD4 and CD8 gates appear to show issues with compensation as the CD8+ population is also CD4-positive. This would indicate potential problems in the staining panel that could lead to data misinterpretation.”
Response: Again, we sincerely appreciate Reviewer 1 thorough examination of the flow cytometry panels. In our turn, we have re-examined the experimental data in this experiment and also conducted an additional experiment to verify previously obtained data. The issue is that in the originally presented data a minor population appears CD4+CD8+. We repeated analysis with FMO (fluorescence-minus-one) controls for both CD4 and CD8, and actually, minimal spillover was found with the FMO controls, as less than 3% of CD8+ cells were CD4+. The vast majority of CD8+ cells were CD4- (97% purity). This negligible spillover effect does not significantly impact the overall data interpretation.
Moreover we can postulate a biological interpretation, which is that the population in the upper right quadrant (Q2) represents true CD4+CD8+ double-positive T cells (a known transitional subset).
About the relevance to main findings presented in the article: the CAR+ cell subset analyses presented in the article were obtained through separate experiments with simplified staining panels (fewer fluorochromes in the staining mixture, gating strategy detailed in Supplementary Figure S5). In these critical experiments we observed no CAR+CD4+CD8+ triple-positive population. This indicates that the key findings about CAR+ cell subsets presented in the article are robust and unaffected.
Reviewer 1 wrote: “Figure 6 – Please add representative flow plot(s) of the CD45RA+CCR7+ subset gating either in Figure 6 or in an additional supplementary Figure. Since there is a significant difference between V1 and V2 within the CD8 compartment, it is recommended to show representative flow plots of this phenomenon.”
Response: The requested changes were done. Figure 6 in the revised manuscript now includes dot plots from flow cytometer with CD45RA+/CCR7+ staining, to demonstrate the distribution of T-cell subsets (Tn, Tcm, Tem, Temra) in a healthy donor leukapheresis (control), or in our manufactured CAR-T cell products.
Reviewer 1 wrote: “Discussion – Others have published CliniMACS Prodigy-based manufacturing of CAR-T cells, could the authors compare their method / results with other published results?”
Response: The revised manuscript now includes additional text comparing our results with key findings from other researchers who have used the CliniMACS Prodigy system for academic-scale CAR-T cell production. This new text can be found in the Discussion section (lines 653–662).
Reviewer 1 wrote: “Minor comments:
Section 2.5 – these were not healthy donors, but rather unclaimed/unused products collected for treatments. While the authors are unable to access specific patient information, could they perhaps share what types of conditions the center would collect these products for? For example, are these potentially leukemia patients?”
Response: We have obtained the requested demographic and diagnosis information (still anonymous) from the transfusion center that provided the leukapheresis products. The ages, genders, and clinical diagnoses are now available in Supplementary Table S1, which has been added to the revised Supplementary Materials.
Reviewer 1 wrote: “Line 260 – CAR-T expansion. The authors mention a rest step after thaw (day -1 to day 0), but interestingly add IL7 and IL15 cytokines. Adding cytokines (signal 3) without CD3/28 (signals 1 and 2) can promote T cell anergy. In future adaptations of the expansion, it is recommended to withhold cytokines during rest/recovery. In addition, could the authors please add the vessel used for recovery step (i.e., flask, bag, etc).”
Response: We express our much gratitude for this practical recommendation. For future CAR-T cell manufacturing, we will not use IL-7 and IL-15. These technical insights are valuable for us, because of the task to develop clinical-grade CAR-T production in our institution.
Regarding culture vessels: we have added the information on the vessel in lines 245-249 in the revised manuscript. Actually, we used non-coated tissue culture roller bottles of sufficient volume. The leukapheresis material washed from cryoprotectants and resuspended in medium (TexMACS+HSA) was transferred into these bottles and placed in a roller CO2-incubator, with the roller set to the minimum possible rotation speed (0.08 rpm). All technical details are presented in lines 245-249 of the revised manuscript.
Reviewer 1 wrote: “Figure 5a – While the image on Day 3 is helpful, the images presented on Day 5 and Day 12 are not clear, and Day 12 is very dark. This could be due to mixing being “on” and thus there are no T cells present in the image plane, or it could be improper focusing of the microscope. It is recommended to replace these images or delete. In addition the text states “darkly stained aggregates”, but, it is important that there is no stain present and thus it is recommended that “darkly stained” be removed from text.”
Response: We addressed this issue as required. Images on days 5 and 12 were removed. The text about “darkly stained aggregates” was removed completely from the revised manuscript.
Reviewer 1 wrote: “Figure 7 – it would be helpful to graph IFN-g, TNF-a, and IL-2 in separate graphs in order to adjust the axes, so that the reader can more easily visualize the differences between V1 and V2.”
Response: The requested changes were done. Figure 7 in the revised manuscript now has graphs for IFN-g, TNF-a, and IL-2 in separate graphs.
Reviewer 1 wrote: “Section 3.4, Functional Analysis – the authors isolated B cells from the apheresis products instead of using a CD19+ cell line. This could present potential issues due to donor variability of B cell quality, especially for example comparing V1 vs V2 groups. Could the authors explain the rationale? Please also state in the methods if B cells were isolated from all 12 products, or from a subset of products.”
Response: This is an important comment regarding the study design. We fully acknowledge that donor variability (here, in B cell properties) may theoretically influence the results. However, our investigation was designed specifically to address this question: whether such variability would permit the reliable use of donor-derived B cells as targets in CAR-T cell cytotoxicity test. The obtained answer is: yes. We show that CAR-T cells effectively target and kill autologous B cells in vitro. Our rationale for using primary B cells was threefold: 1. to provide for better simulating the in vivo hematology tumor; 2. to develop a robust cytotoxicity assay using biologically relevant targets in the form of primary B cells (and, in future, isolated B-blasts); 3. to have more data in hands to convince our collaborating clinicians that CAR-T cells work not only in killing of a specific (immortalized) cell line, but are efficient killers of antilogous CD19+ targets.
We demonstrate in the article that using patient-derived B cells as targets is as effective as the CD19+ cell line. We believe this aspect only enhances the scientific value of the presented work.

Reviewer 2 Report
Comments and Suggestions for Authors
The manuscript presents the first academic-scale CAR-T cell manufacturing platform in Kazakhstan and Central Asia. Deploying both in-house lentiviral vector production and automated manufacturing pipelines in a region without prior experience in cell and gene therapy is both novel and of clear added value.
The work addresses critical access gaps for advanced immunotherapies in low- and middle-income settings and provides a practical, adaptable framework for other developing countries.
By demonstrating that high-quality CAR-T products can be manufactured with locally sourced reagents and existing automation technology, the manuscript substantially advances the global field of cellular immunotherapy.
The documentation of technical, regulatory, and operational aspects adds significant real-world utility. Sharing the exact sequences of the CAR constructs is essential for reproducibility, transparency, and broader utility to the field and should be a general rule. Precise sequences allow others to understand, verify, and, if needed, replicate or adapt the approach, especially given the subtle but important role that small sequence diferencies may play in CAR-T cell biology.
The manuscript is well-structured and clearly written, with thorough methods and results sections. Diagrams and tables effectively illustrate workflow, production metrics, and results.
Figures provide visual support for the presented findings, although some legends could be more explicit regarding sample sizes or the specifics of experimental controls.
The abstract, introduction, and discussion sections are cohesive and informative.
The experimental design and analyses are appropriate and executed to a high standard. In-house lentiviral vector production and the use of an automated, GMP-compatible system support the robustness of the approach.
Phenotypic, functional, and statistical analyses are comprehensive and support the authors' claims on product quality, functional activity, and comparability to reference standards.
However, there is a notable limitation in the design: when comparing the two CAR constructs (Kymriah-like and Yescarta-like), it must be emphasized that the constructs differ not only in the costimulatory domain (4-1BB vs. CD28), but also in the hinge and transmembrane regions. This affects the interpretation of functional differences, as observed effects could be attributed to structural differences beyond just the costimulatory domain. The authors should clearly articulate this caveat when discussing the comparative data.
The manuscript will be of interest to researchers and clinicians in cell therapy in resource-limited settings, and global health professionals interested in translational medicine and technology adaptation.
The data and workflow presented are especially valuable for those looking to establish similar platforms or leverage academic production models.
This is a timely manuscript that provides knowledge and fills practice gap in academic CAR-T cell production outside established centers.
The study is likely to serve as a reference point for both the technical and policy aspects of implementing advanced cell therapies in new markets.
Key recommendations for the authors:
Clearly discuss the implications of comparing CARs that differ in multiple structural domains, not just costimulatory regions, and urge caution in attributing functional differences solely to the costimulatory moiety.
Consider expanding on logistical or regulatory lessons learned for maximal global impact.
In summary, I recommend the manuscript for publication pending these essential clarifications and additions.
Author Response
Reviewer 2 wrote: “Comments and Suggestions for Authors:
The manuscript presents the first academic-scale CAR-T manufacturing…in Kazakhstan and Central Asia...clear added value.
Sharing the exact sequences of the CAR constructs is essential for… broader utility in the field and should be a general rule.
The manuscript is well-structured and clearly written.
The experimental design and analyses are appropriate.”
Response: This is really generous assessment of our work's significance and quality, for which we express gratitude to Reviewer 2. These affirming words encourage the authors to continue towards the first clinical application of CAR-T therapy in Kazakhstan. And to date, our group and institution remain the only in Kazakhstan capable of manufacturing of clinical-grade CAR-T cell products, and having established collaborations with the clinic to enable the transition into clinic.
Reviewer 2 wrote: “However, there is a notable limitation in the design: when comparing the two CAR constructs (Kymriah-like and Yescarta-like), it must be emphasized that the constructs differ not only in the costimulatory domain (4-1BB vs. CD28), but also in the hinge and transmembrane regions. This affects the interpretation of functional differences, as observed effects could be attributed to structural differences beyond just the costimulatory domain. The authors should clearly articulate this caveat when discussing the comparative data.”
Response: We appreciate this accurate comment. As suggested by the reviewer, we have added clarifying text in the Discussion (lines 624-634) explicitly stating that the functional differences may stem from combined effects of both costimulatory domains and structural disparities (hinge/transmembrane regions) between the two CAR constructs.
Reviewer 2 wrote: “The manuscript will be of interest to researchers and clinicians in cell therapy in resource-limited settings…especially valuable for those looking to establish similar platforms…academic production.
The study…to serve as a reference point for…implementing cell therapies in new markets.”
Response: We thank Reviewer 2. The submitted article serves purposes that extend beyond a conventional scientific report. Yes, we want to facilitate wider dissemination of components of the Cell and Gene Therapy (CGT) such as lentiviral vector production, and genetically modified cell therapy production (having the CAR-T as an example), in developing word. Also we want to advertise our presence in the field, both in Kazakhstan and abroad, to have more contacts from clinical community in our country, and more international collaboration with foreign parties, possibly with CAR-T startups from developed countries which want to have a new ground for implementing their technologies. Regulations of implementing advanced therapies in Kazakhstan is easier than that of FDA or EMA, so there may be motivated interest in some of competent developers, including startups, to collaborate with us to promote (their) technologies.
Reviewer 2 wrote: “Key recommendations for the authors:
Clearly discuss the implications of comparing CARs that differ in multiple structural domains, not just costimulatory regions, and urge caution in attributing functional differences solely to the costimulatory moiety.”
Response: As suggested by the reviewer, we have added an additional paragraph to the Discussion, in lines 624-634. We discuss the functional differences may stem from combined effects of both costimulatory domains and structural disparities (hinge/transmembrane regions) between the two CAR constructs.
Reviewer 2 wrote: “Consider expanding on logistical or regulatory lessons learned for maximal global impact.”
Response: Indeed, through this project, we identified that implementing CAR-T therapy in a country without prior experience in gene and cell therapies requires addressing numerous challenges, including solving logistics and regulatory problems. We have added description of our experience in the Discussion section, considerations on the logistics in lines 565-568; and challenges we have faced with the national regulator in lines 683-689.
Reviewer 2 wrote: “In summary, I recommend the manuscript for publication pending these essential clarifications and additions.”
Response: Thank you!
